# The Scientific Productivity of Collective Subjects Based on the Time-Weighted PageRank Method with Citation Intensity

Alexander Kuchansky [1,2], Andrii Biloshchytskyi [3,4,*], Yurii Andrashko [5], Svitlana Biloshchytska [4,6] and Adil Faizullin [7]

1   Department of Information Systems and Technology, Taras Shevchenko National University of Kyiv, 01601 Kyiv, Ukraine
2   Department of Cybersecurity and Computer Engineering, Kyiv National University of Construction and Architecture, 03680 Kyiv, Ukraine
3   Administration, Astana IT University, Astana 010000, Kazakhstan
4   Department of Information Technology, Kyiv National University of Construction and Architecture, 03680 Kyiv, Ukraine
5   Department of System Analysis and Optimization Theory, Uzhhorod National University, 88000 Uzhhorod, Ukraine
6   Department of Computer Technology and Data, Astana IT University, Astana 010000, Kazakhstan
7   Department of Quality Assurance, Astana IT University, Astana 010000, Kazakhstan
*   Correspondence: bao1978@gmail.com

**Abstract:** This study aims to estimate the scientific productivity of collective subjects. The objective is to build a method for evaluating scientific productivity through calculation, including for new collective subjects with a small citation network—the paper proposes the Time-Weighted PageRank method with citation intensity (TWPR-CI). The Citation Network Dataset (ver. 13) has been analyzed to verify the method. The dataset includes more than 5 million scientific publications and 48 million citations. Four classes of collective subjects (more than 27,000 collective subjects in total) were established. For each class, scientific productivity estimates from 2000 to 2021 were calculated using the PageRank, Time-Weighted PageRank, and TWPR-CI methods. It is shown that the advantage of the TWPR-CI method is the higher sensitivity of the scientific productivity estimates for new collective subjects on average during the first ten years of observation. At the same time, the assessment of scientific productivity for other collective subjects according to this method is stable. However, the small citation network of the new collective subjects prevents the adequate assessment of scientific productivity during the first years of its operation. Therefore, the TWPR-CI method can be used to assess the scientific productivity of collective subjects, in particular the productivity of new ones.

**Keywords:** PageRank; Time-Weighted PageRank; collective subjects; citation intensity; scientific research; research productivity; scientometrics

## 1. Introduction

The development and use of methods for evaluating the scientific productivity of collective subjects was and remains an urgent task in scientometrics. There are different methods of evaluating the productivity of individual subjects (scientists) and collective subjects (higher education institutions, scientific institutions, faculties, departments, etc.). The key to forming the reputation of any academic educational institution or collective subject is the scientific productivity of its employees. Most of the known indices of evaluation of collective subjects are based on the productivity of the scientific activity of employees affiliated with them. The productivity of scientific activity should be calculated based on quantitative scientometric indicators determined transparently and independently of subjective factors, primarily using open sources.

Common approaches for calculating scientometric performance indicators of collective subjects frequently use traditional citation indices such as the h-index [1]. The h-index

determines an author's influence or productivity based on the number of citations to one's scientific publications. However, when calculating scientific productivity, the h-index, and its analogs, such as the i10-index, g-index, etc., lose some citations placed outside the core. Modern methods of analyzing citation networks can consider information about all citations of an author's network.

The emergence of the PageRank method [2] offered new possibilities for evaluating collective subjects' scientific productivity and reputation. The traditional purpose of the PageRank (PR) method is to establish users' influence in social networks or evaluate web pages' importance. Each network user or page is assigned a valid number representing importance or reputation. The larger this number, the greater the importance [3]. There are many modifications of the PR method used to calculate scientific productivity, citation index and the reputation of scientific journals, etc.

The assessment of collective subjects is based on the principle that this appraisal is the convolution of the estimates of scientific productivity of all scientists affiliated with a particular collective subject. In [4], it is shown that if the growth potential of scientific productivity estimates for individual subjects is positive, then the potential of the collective subject will be positive as well. Moreover, individual subjects are affiliated with the collective subject.

The dynamic development of academic space should be considered to assess the scientific productivity of collective subjects. Relying on calculating productivity scores based on classical citation indices is inappropriate. This is because such calculation methods are limited to the core of the quotes. On the other hand, calculating the assessment of the scientific productivity of collective subjects by taking into account all the citations of scientists (the classic PR method) is also questionable. In particular, it is because the larger the network of citations, the greater the probability of its rapid growth. It can be assumed that the indicated methods will not provide reliable evaluation results for new universities and scientific institutions. Even if university employees have a higher publishing and scientific activity dynamic, a history of citations of publications of sufficient volume is required to obtain a reliable assessment of a collective subject's scientific productivity. Evaluation of scientific productivity based on the classic PR method will be delayed in time. The classical PR method evaluates the citation network as a static object and does not take into account the intensity of citations. The number of citations of scientific publications recorded over a specific period of time is not considered when calculating the assessment of scientific productivity.

Therefore, it can be assumed that the classical PR method is well suited for assessing individual subjects and is generally not suitable for assessing the scientific productivity of collective subjects, particularly new ones. This can at least be said for the indicated method in the traditional interpretation. That is why it is considered essential to develop a modification of the PR method for evaluating the scientific productivity of collective subjects, considering the intensity and aging coefficient of citations of scientific publications by authors affiliated with the collective subject. The study's results will theoretically and practically enrich scientometrics' area in evaluating scientific productivity for universities, scientific institutions, and structural divisions of these institutions. Therefore, research on developing a modification of the PR method considering the intensity and age of citations is relevant.

The purpose of the study is to construct the Time-Weighted PageRank method with citation intensity (TWPR-CI) for evaluating the scientific productivity of collective subjects. The Time-Weighted PageRank method with citation intensity considers the age and intensity of citations of scientific publications by authors affiliated with collective subjects. Research hypothesis: using the modified TWPR-CI method increases the sensitivity of scientific productivity assessment compared to the classic PageRank and the TWPR methods. This allows for the adjustment of the position of new collective subjects in terms of the rating of scientific productivity. In the case of using the classic PR method and the TWPR method for evaluating scientific productivity, preference is given to long-standing collective

subjects, and articles affiliated with them have a sufficient volume of citations. Despite the insufficient volume of the citation network of scientific publications, the use of citation intensity corrects the assessment of scientific productivity according to the TWPR method for new collective subjects.

The classical PR method uses only edge relations and does not consider higher-order structures, particularly subgraphs. One of the concepts of modifying the PR method described in [5] is the inclusion of higher-order structures in the calculation. The research in [5] demonstrates that this approach improves the ranking of social network users. This approach makes sense because citation networks tend to have a complex structure. This fact can be used to evaluate scientific productivity effectively. However, it is not easy to use this method in real-time. A dynamic change in the structure of the citation network leads to the need for a new recalculation of scientific productivity estimates. Moreover, this calculation is cumbersome. In [6], an iterative method for calculating PR is proposed, simplifying the calculation of scientific productivity estimates to a certain extent.

One of the areas of scientometrics to which the class of PR methods is actively applied is the ranking of scientific journals. In particular, this applies to the well-known impact indices of SCImago Journal Ranking [6] or EigenFactor article impact assessments [7]. In [8], a weighted PageRank method is proposed, considering the h-indexes of journal authors. Experimental results show that the HR- PageRank method proposed in [9] outperforms the well-known PR method in finding influential journals according to statistical evaluation data. The HR-PageRank evaluation can be used to assess the scientific productivity of collective subjects. However, part of this hybrid method is the well-known h-index, the disadvantage of which is the rejection of authors' citations outside the calculation core.

One of the first attempts to use the PR method to calculate the productivity assessment of collective subjects of scientific activity was carried out in [10]. In [10], the rating of collective subjects is based on the results of the PR assessment of 24 articles in the Wikipedia publication. Comparing the rating calculated for the top 100 universities with the ARWU-500 list in [10] revealed that the ratings coincide by 62%. This indicates that the analysis, in general, produces reliable results. However, the specified rating does not consider the evaluation of scientific productivity and the results of the publication activity of authors affiliated with the universities included in this rating.

Another concept that can be used in calculating estimates of the scientific productivity of collective subjects is assessing the impact of textbooks published by them. In particular, [11] analyzed 1869 textbooks from the funds of indexed books in the Scopus database. The descriptive statistics method shows the relationship between the teaching ranks of textbooks used in world-class universities according to the Times Higher Education tool and indicators obtained from citations using the PR method. However, the publication of manuals only reflects part of the scientific activity of the scientific team. Therefore, it is necessary to consider all types of scientific activity in the complex to obtain an adequate assessment of scientific productivity. In [12], the 108 most cited authors in the field of information retrieval (IR) from the 1970s to 2008 were studied. The analysis made it possible to form a network of joint citations. It is shown that the growth of the author's influence, determined by the citation of one's scientific publications, affects the growth of the PageRank rating.

In [13], five different Web of Science scientific areas were investigated for assessing academic reputation based on the PageRank method. These areas correspond to research topics studied according to recognized international academic classifications. In [14], an optimized PageRank method is proposed using the Labeled Latent Dirichlet Allocation (Labeled-LDA) thematic model. The indicated areas are relevant in estimating scientific productivity within the corresponding scientific area. These areas can be identified based on the corresponding thematic model. However, in each scientific area, there are peculiarities related to the intensity of citations of publications and the appearance of new studies. Accordingly, the assessment of the scientific productivity of a collective subject can be based on the productivity in the corresponding scientific area to which the respective

collective subject belongs to a certain extent. However, without considering the authors' age and citation intensity, it is difficult to determine an adequate assessment of the scientific productivity of the corresponding collective subject with which these authors are affiliated.

In [15], citation networks are considered, and a characteristic of aging of citations is introduced in the PR method, considering only 10-year citations. The study's results indicate that considering the aging characteristics improves the performance of the PR algorithm. The limitation of the time of appearance of a scientific publication and citations to it, similar to the h-index, limits the possibility of selecting promising collective subjects. A proposal that can objectively improve the consideration of citation aging for evaluating the scientific productivity of collective subjects is not to limit the term of their appearance but to take them into account with an appropriate aging factor. In [16], it is shown that assigning weights to the edges in authors' collaboration network, according to a decreasing exponential function depending on the time elapsed since the publication of a common paper, may add valuable information to the process of ranking authors based on importance. The research in [17] describes the weighted algorithm PageRank algorithm (WPR). Its advantage over the standard PR algorithm is shown. The concept of PR weighting by time is described in [18].

An essential tool for evaluating scientific productivity dynamics is considering the citations' intensity. This allows us to state that it is appropriate to conduct a study devoted to developing a modified PageRank method of evaluating the scientific productivity of collective subjects, considering the age and intensity of the citation of scientific publications.

The following tasks were outlined to achieve the goal:

- Description of components used in the combined Time-Weighted PageRank method with citation intensity. Solving the problems of calculating the intensity of scientific publications citations and the problem of weighting the coefficients of the PageRank method over time;
- Description of Time-Weighted application possibilities of the PageRank method with citation intensity for evaluating the scientific productivity of collective subjects.

## 2. Methods and Data

### 2.1. Basic Terms and Concepts

Some terms and concepts have been used in the publication. The intensity of scientific publication citation is the speed of change in the number of citations of some subjects' publications. The sensitivity is the speed of change in the scientific productivity assessment.

Individual subjects are scientists. Each scientist is affiliated with some collective subject. Collective subjects should be higher education institutions, scientific institutions, faculties, departments, etc.

The productivity of scientific activity is a relative value calculated on the bases of quantitative scientometric indicators and determined transparently and regardless of subjective factors, primarily using open sources.

### 2.2. The Assessment of Scientific Productivity

The research assumes that the modified TWPR-CI method increases the sensitivity of the assessment of the scientific productivity of new collective subjects compared with the PR method and the TWPR method. The modified method should solve the issue of evaluating the scientific productivity of new collective subjects at the stage of forming their citation networks. The insufficient volume of citation data in new collective subjects leads to an underestimation of their scientific productivity by the usual PR method. This is embedded in the structure of the PR calculation, despite the high scientific activity of authors from new collective subjects. Subsequently, with the citation network's growth, scientific productivity assessment stabilizes, and the volume of the network does not affect the result. This feature of the PR method for calculating scientific productivity can be corrected by introducing a weighting factor for the parameters of the method by time and a citation intensity factor.

To build the modified PageRank method (TWPR-CI), taking into account the age and intensity of citations, a class of algorithms for evaluating the importance of web pages based on solving a system of linear algebraic equations was used. The traditional iterative Gauss–Seidel or Liebmann method was used to solve systems of equations. Filtering with a finite impulse response (FIR), in particular, the principle of calculating the linear weighted moving average (LWMA), was used to calculate the coefficient that determines the citation age. Furthermore, the equation for finding the angular coefficient of the straight line, which reflects the intensity of change in the number of citations of a specific scientific publication, was used to consider the intensity of citations. Finally, the graph construction method was applied, the vertices of which are scientific publications, and the arcs are citations of one publication in others.

A dataset of scientific publications, Citation Network Dataset (ver. 13) [19], was analyzed to verify the research. The details of its construction are described in [20]. This set contains data on 5,354,309 scientific publications and 48,227,950 citations of these publications, collected from databases DBLP [21], ACM [22], Microsoft Academic Graph [23], and others. The specified version contains current data on publication citations as of May 2021.

If U = {$U_1$, $U_2$, ..., $U_s$} is the set of collective subjects, A = {$a_1$, $a_2$, ..., $a_d$} is the set of individual subjects. Certain individual subjects are affiliated with each collective subject:

$$f : A \times U \rightarrow \{0, 1\}. \tag{1}$$

Let us denote $A^h = \left\{ a_1^h, a_2^h, \ldots, a_{d_h}^h \right\}$ as the set of individual subjects affiliated with the collective subject $U_h$, $h = \overline{1, s}$, $a_j^i \in A$, $A^h \subset A$, $j = \overline{1, d_h}$, $d_h$ is the number of individual subjects affiliated with the corresponding collective subject $U_h$, s is the number of the collective subject. Each individual subject develops in its own information space. The information space includes the history of the publication activity of an individual subject.

Let P = $\{p_1, p_2, \ldots, p_n\}$ be the set of all scientific publications. Each individual subject is associated with a set of scientific publications and citations of these publications, both outgoing and incoming. Let $p^t\left(a_j^h\right)$ be the number of scientific publications published by an individual subject $a_j^h$ at time t, $t \in T$, $T = \{t_1, t_2, \ldots, t_N\}$. Let $c_i^t\left(a_j^h\right)$ be the number of citations of a scientific publication $p_i$ at time t of an individual subject $a_j^h$ affiliated with a collective subject of scientific activity $U_h$, $i = \overline{1, g_j^h}$, $g_j^h$ is the number of scientific publications of an individual subject $a_j^h$.

Furthermore, it is necessary to set the Markov matrix that determines the citation between publications through $M = \left\{ c_{ij} \right\}_{i,j=1}^n$, where n is the total number of scientific publications, $c_{ij} \in [0, 1]$ is the probability of transition from one state to another, i.e., in the context of the PR method is the probability of citing one scientific publication in another, $M \geq 0$, $\sum\limits_{i=1}^n c_{ij} = 1$, $j = \overline{1, n}$.

Let the coefficient that determines the weight of the scientific publication $p_i$, on the basis of which the rank of the publication is calculated at the k-th step, be denoted by $r_i^k$. At the initial stage (step k = 0), the coefficients for all publications are equal and are defined as $r_i^0 = \frac{1}{n}$, $i = \overline{1, n}$. All other coefficients are calculated iteratively according to the following equation:

$$r_i^{k+1} = \alpha M r_i^k + \frac{1-\alpha}{n} E \tag{2}$$

where **E** is the unit matrix, $\alpha$ is the damping factor, which determines the probability of transition from one state (current scientific publication) to another state (another random scientific publication).

The coefficients $r_i^k$ are calculated iteratively, given that after performing a sufficient number of iterations, for $k \rightarrow +\infty$, we will obtain an approximate value of the coefficients

$r_i^k$. In [5,6], an iterative algorithm for calculating coefficients $r_i^k$ according to the following equation is proposed:

$$r_i^{k+1} = \sum_{j=1}^{i-1} r_j^{k+1} c_{ji} + \sum_{j=1}^{n} r_j^k c_{ji} \tag{3}$$

The stop condition is the fulfillment of the inequality $\left| r_i^{k+1} - r_i^k \right| < \varepsilon$ for a fixed small value of $\varepsilon > 0$. As a result of the calculations, we obtain the vector $(\tilde{r}_1, \tilde{r}_2, \ldots, \tilde{r}_n)$, where $\tilde{r}_i$ is the calculated value of the coefficient for the scientific publication $p_i$, $\tilde{r}_i \in [0,1]$, $i = \overline{1,n}$, $\sum_{i=1}^{n} \tilde{r}_i = 1$. The publication $p_i$ corresponding to the maximum value of the coefficient $\tilde{r}_i$ will have the rank $R(\tilde{r}_i) = 1$. The ranks of other publications are calculated in order of decreasing coefficients $\tilde{r}_i$, $i = \overline{1,n}$. We can then obtain the ranks of scientific publications as follows:

$$(R(\tilde{r}_1), R(\tilde{r}_2), \ldots, R(\tilde{r}_n)), \ R(\tilde{r}_i) \in \mathbb{N}, \ R(\tilde{r}_i) \leq n. \tag{4}$$

If the time of publication of a scientific publication $p_i$ is $t = t_z$, then the number of citations of scientific publication $p_i$ is determined by the vector

$$\left( 0, 0, \ldots, 0, c_i^{t_z}\left(a_j^h\right), c_i^{t_{z+1}}\left(a_j^h\right), \ldots, c_i^T\left(a_j^h\right) \right) \tag{5}$$

A conclusion can be drawn regarding assessing the scientific productivity of a collective subject based on the ranks of scientific publications of affiliated authors. The PR method is considered for calculating the scientific productivity of collective subjects by immediately setting the appropriate scores in the coefficients of the PR method. To accomplish this, Equation (2) is used; however, the coefficients in this equation will determine the scientific productivity of the collective subject rather than the weight of publications. This approach to calculating PR coefficients was used in [24–26].

The matrix of citations between scientific publications of different collective subjects is given as $\overline{M} = \left\{ \overline{c}_{hg} \right\}_{h,g=1}^{s}$, where s is the total number of collective subjects, $\overline{M} \geq 0$, $\sum_{h=1}^{s} \overline{c}_{hg} = 1$, $g = \overline{1,s}$. $q_h^k$ is the coefficient that determines the scientific productivity of the collective subject $U_h$ at the k-th step. For k = 0, we have the coefficients $q_h^0 = \frac{1}{s}$. All other coefficients are calculated iteratively according to the equation:

$$q_h^{k+1} = \alpha \overline{M} q_h^k + \frac{1-\alpha}{s} E \tag{6}$$

where **E** is the unit matrix, $\alpha$ is the damping factor, $h = \overline{1,s}$.

Consider a modified method of calculating scientific productivity, considering the intensity and age of citations. This requires the entry of the citation intensity. The intensity of citation will be determined by the angle coefficient of the straight line drawn between two points that determine the number of citations of scientific publications of the collective subject at the moment $t_\delta^h$ and the number of citations of publications of the collective subject at the current time $t_N$. For this, it is necessary to calculate the value for the collective subject $U_h$:

$$\theta_{t_\delta^h} = \arctan\left( \frac{\sum\limits_{j=1}^{d_h^t} c^t\left(a_j^h\right)}{\lambda\left(t_N - t_\delta^h\right) \sum\limits_{j=1}^{d_h^t} p^t\left(a_j^h\right)} \right) \tag{7}$$

where $\theta_{t_\delta^h}$ is the intensity of citation of scientific publications by authors who are affiliated with the collective subject $U_h$ at the moment of time t, $t \in T$, $d_h^t$ is the number of individual subjects affiliated with the collective subject $U_h$ at the moment of time t, $p^t\left(a_j^h\right)$ is the

number of scientific publications published by the individual subject $a_j^h$ at the moment of time t, t $\in$ T, $c^t\left(a_j^h\right)$ is the number of citations of the scientific publications of the authors, who are affiliated with the collective subject $U_h$ at the moment of time t, $\lambda$ is the parameter, and $\lambda > 1$, $t_\delta^h$ is the moment from which the calculation of the intensity of citations of scientific publications for the collective subject $U_h$ begins. The angle $\theta_{t_\delta^h}$ tangent is equal to the time derivative of the scientific productivity function at the time moment $t_\delta^h$. Additionally, the approximate value of the derivative can be calculated as an argument of the function arctan (7). For new collective subjects, the number of scientific publications and citation indicators is zero until the first publications of authors affiliated with them appear.

The linear coefficient of aging of scientific publications of a collective subject over time is introduced. The closer the time of publication to the current point in time $t_N$, the greater the influence of citations of these publications on the evaluation of its PR. Papers published a long time ago will have a lower coefficient, and their influence on the result of the scientific productivity ranking will be reduced. The coefficient $q_h^0$ is determined by taking into account the age and intensity of citations as follows:

$$q_h^0 = \beta \cdot \theta_{t_\delta^h} + (1 - \beta) \cdot \sum_{k=t_\delta^h}^{N} \frac{\left(k - t_\delta^h + 1\right) \cdot c^{t_k}\left(a_j^h\right)}{x_i} \qquad (8)$$

where $x_i = \sum_{k=t_\delta^h}^{T_N} \left(k - t_\delta^h + 1\right)$, $q_h^0$ is the value of the coefficient taking into account the age and intensity of citations for the collective subject, $U_h$, $h = \overline{1, s}$, $\beta \in [0, 1]$.

A modification of the method by which the coefficient is calculated according to Equation (8) is called the TWPR-CI method with the $\beta$ parameter. If $\beta = 0$, then we adopt the TWPR method. The collective subject $U_h$, which corresponds to the maximum value $q_h^k$ at the k-th step, will have the $R\left(q_h^k\right) = 1$ rank, etc., in order of decreasing value of the coefficient $q_h^k$. The maximum rank of a collective subject corresponds to the maximum scientific productivity of this collective subject. The value of k is determined by taking into account the stop condition according to the iterative PR method. As a result of calculations at the k-th step, we obtain a vector of coefficients $(\widetilde{q}_1, \widetilde{q}_2, \ldots, \widetilde{q}_s)$. The result of the TWPR-CI method is a ranked list of collective subjects according to the criterion of maximum scientific productivity.

$$\left(R(\widetilde{q}_1), R(\widetilde{q}_2), \ldots, R(\widetilde{q}_s)\right), \; R(\widetilde{q}_h) \in \mathbb{N}, \; R(\widetilde{q}_h) \le s, \; h = \overline{1, s}, \qquad (9)$$

where $R(\widetilde{q}_h)$ is the rank of the collective subject $U_h$, $\widetilde{q}_h$ is the scientific productivity of the collective subject $U_h$, $h = \overline{1, s}$.

## 3. Results

### 3.1. Collection of Data on Citations of Scientific Publications of Collective Subjects

The Citation Network Dataset (ver. 13) [19] of scientific publications was analyzed. This dataset contains information on 5,354,309 scientific publications as well as 48,227,950 citations to these publications. The data were collected from the DBLP, ACM, and Microsoft Academic Graph databases. The specified version of the dataset includes current data on the citation of scientific publications as of May 2021. The dataset contains scientific publications for the period from 1815 to 2021. However, the publications are unevenly distributed over time. About 87% of the scientific publications in the dataset were published between 2000 and 2021.

Based on the affiliation of the authors of scientific publications, 27,500 unique collective subjects were identified. Collective subjects are mainly institutions of higher education and research institutions, as well as individual private companies and separate structural

divisions of universities. Most publications belong to the following areas: computer science, artificial intelligence and artificial neural networks, mathematics, combinatorics, and software engineering.

### 3.2. The Results of the Calculation of Estimates of the Scientific Productivity of Collective Subjects

The Citation Network Dataset is used to calculate binary mappings between scientific publications and collective subjects. A graph of citations of scientific publications of some collective subjects in scientific publications of other collective subjects was constructed. The citation graph is a directed weighted graph, and the weight of the arcs of the citation graph is equal to the number of citations from scientific publications. Figure 1 shows a part of the citation graph, which includes 30 selected collective subjects. The citation graph is built using open-source and multiplatform software Gephi 0.9.7 [27].

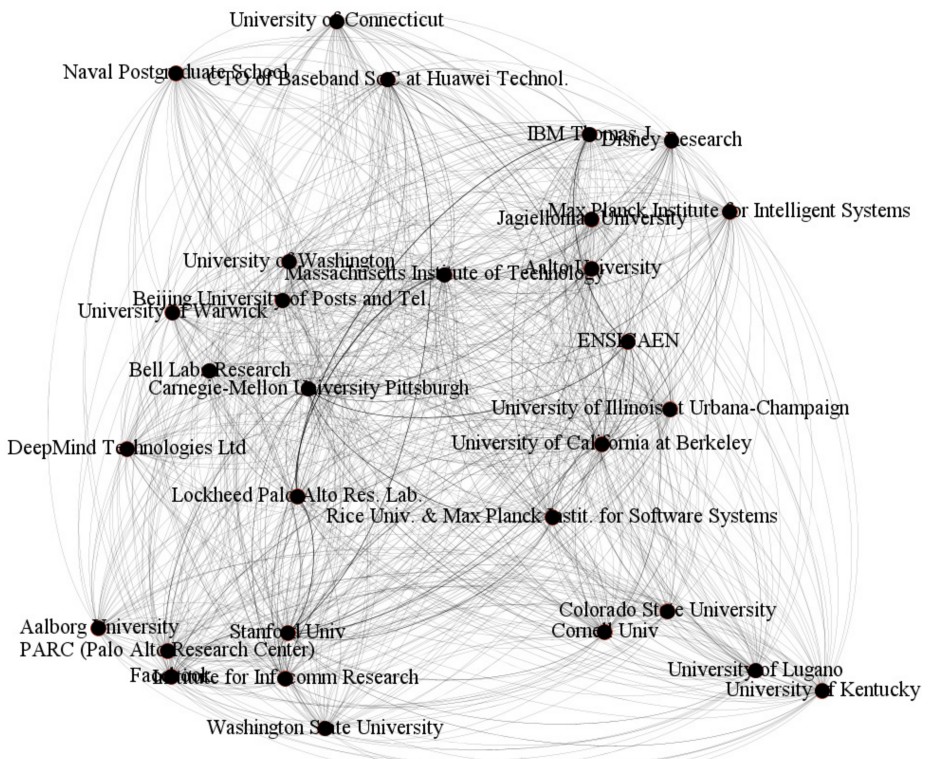

**Figure 1.** Graph of citations of scientific publications for 30 selected collective subjects.

Based on the obtained graph of citations, for calculating estimates of the scientific productivity of collective subjects methods of PR (6), TWPR (6), (8) for $\beta = 0$ and TWPR-CI (6), (8) for $\beta = \frac{1}{2}$ have been used. The value of the intensities of scientific publication citations of collective subjects (7) was calculated as well.

Since the vast majority of scientific publications in the Citation Network Dataset were published after 2000, the period from 2000 to 2021 in one-year increments was chosen to study the dynamics of estimate changes. Accordingly, 21 scientific productivity estimates were calculated for each of the collective subjects using three methods (PR, TWPR, TWPR-CI). In addition, the intensity of the publication citations of each was calculated. Estimated scores for various collective subjects are given in Appendix A. The score of a collective subject for a given year includes only those scientific publications dated until 31 December of the corresponding year.

Estimates of scientific productivity are found based on an iterative method with an accuracy of $\varepsilon = 10^{-5}$. Furthermore, the estimates of scientific productivity were normalized with the maximum value. Accordingly, all estimates of the scientific productivity of collective subjects belong to the interval [0, 1].

During the calculations, it was found that the collective subjects from the dataset Citation Network Dataset were divided into several classes with similar properties. Accordingly, all collective subjects were divided into the following four classes: new collective subjects (N), well-known collective subjects (WK), non-cited collective subjects (NC), and other collective subjects (O) (Table 1). The class of non-cited collective subjects (NC) includes those collective subjects whose publications have never been cited. According to the properties of the PR method and its modifications, the assessment of scientific productivity for the class of non-cited collective subjects (NC) is equal to 0. Therefore, in further research, collective subjects of the NC class are not considered.

**Table 1.** Distribution of collective subjects by classes.

| Class | Number of Collective Subjects |
|-------|-------------------------------|
| N | 16,544 |
| WK | 236 |
| O | 4928 |
| NC | 5791 |

The other three classes (N, WK, O) include collective subjects whose publications are cited at least once during the observation period (2000–2021). Class N includes those collective subjects whose publications were all published after 1 January 2001. The WK class includes collective subjects with an assessment of scientific productivity greater than 0.05 for the entire period. The assessment of scientific productivity is calculated according to the PR method. Class O includes all collective subjects that are not included in classes N, WK, and NC.

The average value of citation intensity was calculated for each of the collective subjects' three classes (N, WK, O). Figure 2 shows changes in the average citation intensity of scientific publications belonging to collective subjects from classes N, WK, and O. It can be concluded that despite the difference in the absolute values of the citation intensity, it tends to grow throughout the observation period.

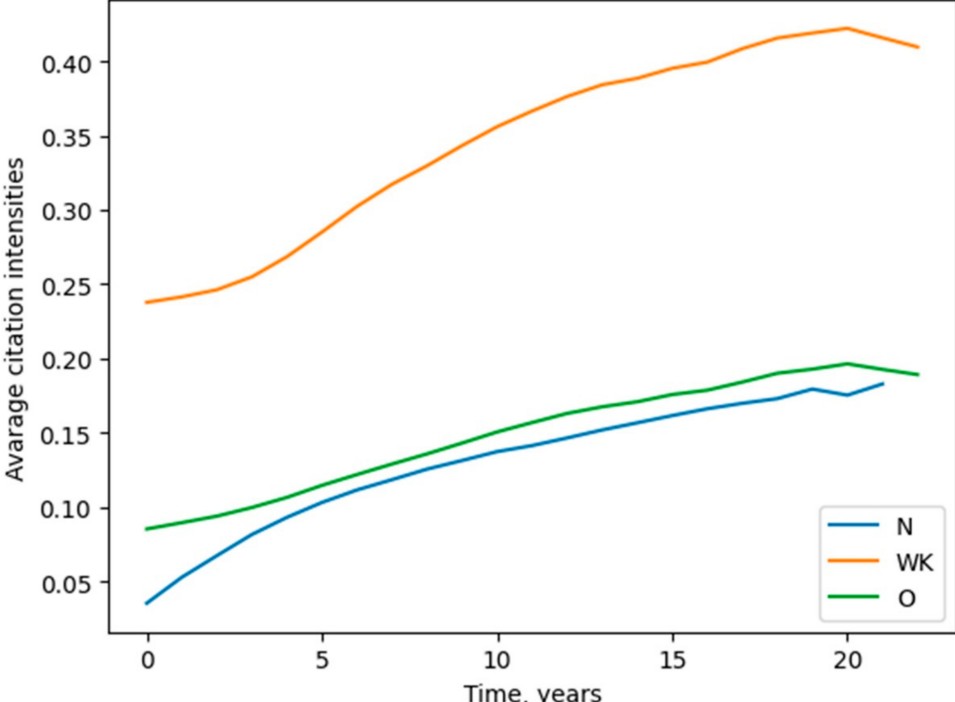

**Figure 2.** Comparison of average citation intensities of collective subjects belonging to classes N, WK, O, $\lambda = 2$.

Comparing the scientific productivity estimates of new collective subjects (collective subjects from class N) obtained with the PR, TWPR, and TWPR-CI methods was essential.

Figure 3 shows changes in the average assessment of the scientific productivity of collective subjects of class N, calculated using the PR, TWPR, and TWPR-CI methods for the period from 2000 to 2021.

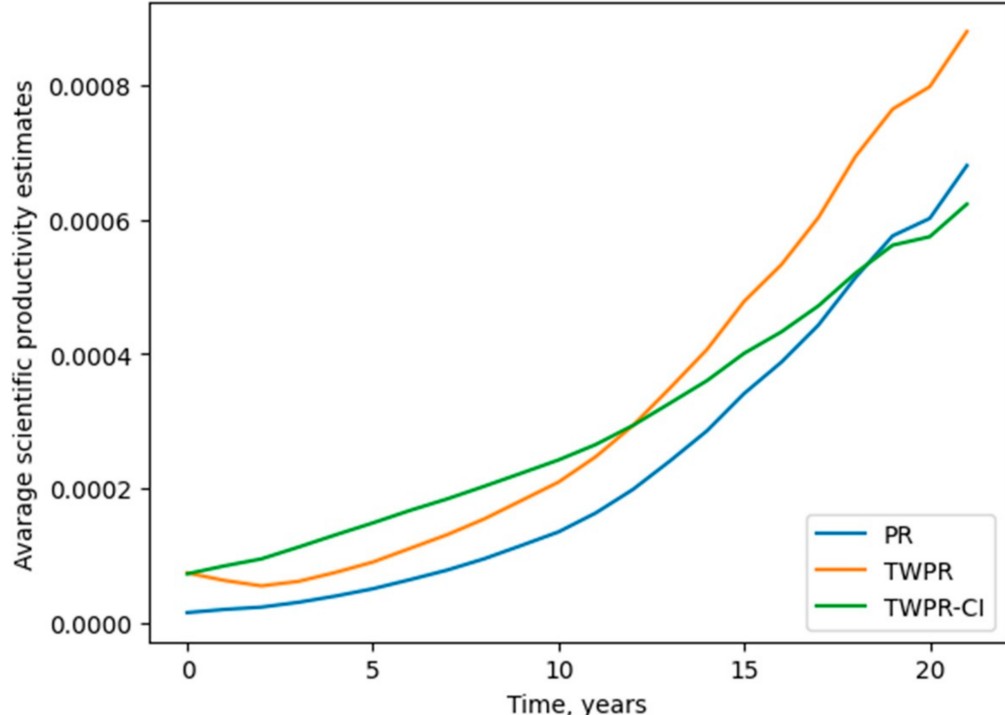

**Figure 3.** Graph of comparison of scientific productivity estimates of collective subjects belonging to class N according to PR (6), TWPR (6), (8), $\beta = 0$, TWPR-CI (6), (8), $\beta = \frac{1}{2}$, $\lambda = 2$.

As can be seen from Figure 3, estimates of the scientific productivity of collective subjects of class N, which are calculated by the TWPR-CI method (6), (8) for the parameter $\beta = \frac{1}{2}$, have greater values than the scientific productivity estimates obtained by the TWPR methods (6), (8) for parameter $\beta = 0$ and PR (6) during the first 12 years (for the dataset Citation Network Dataset). Starting from the twelfth year of observations, the values of the assessment of scientific productivity according to the TWPR-CI method increases more slowly than the assessments of scientific productivity according to the TWPR method.

Figure 4 shows changes in the average assessment of the scientific productivity of collective subjects of the WK class, calculated using the PR, TWPR, TWPR-CI methods for the observation period (from 2000 to 2021).

As can be seen from Figure 4, estimates of the scientific productivity of collective subjects of the WK class, which are calculated by the TWPR-CI method (6), (8) for the parameter $\beta = \frac{1}{2}$, have mostly lower values than the estimates by the TWPR method and higher than estimates by the PR method.

Figure 5 shows the dynamics of changes in the scientific productivity average assessment of the collective subjects of class O, calculated using the PR, TWPR, TWPR-CI methods over the observation period (from 2000 to 2021).

Figure 5 shows the superiority of estimates of scientific productivity by the TWPR method for the entire period of observation for collective subjects of class O.

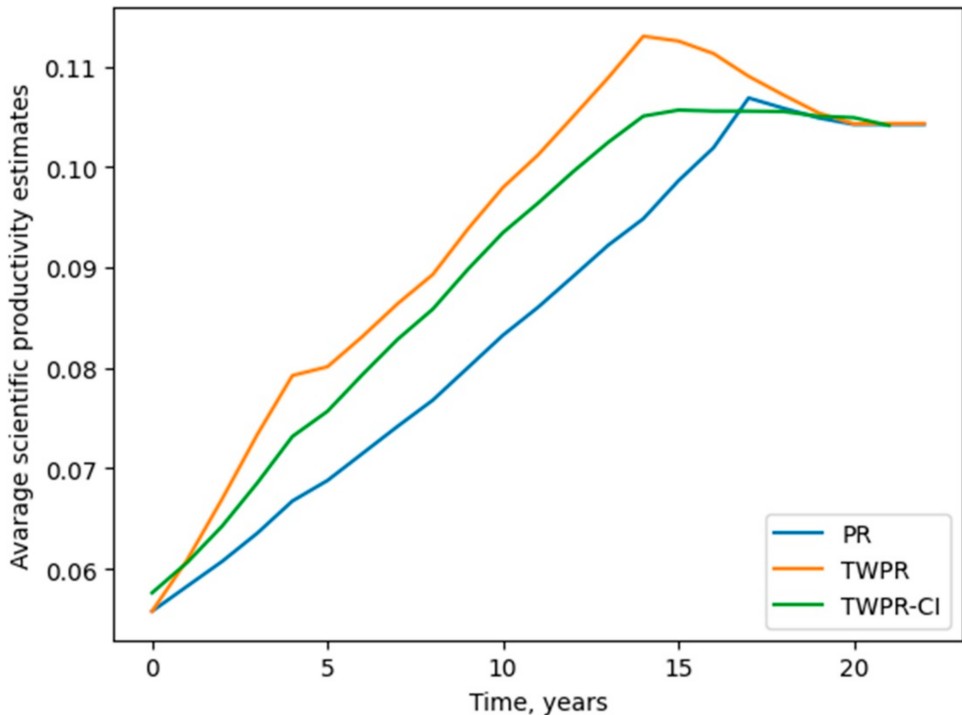

**Figure 4.** A graph comparing the scientific productivity estimates of collective subjects belonging to the WK class by the methods of PR (6), TWPR (6), (8), $\beta = 0$, TWPR-CI (6), (8), $\beta = \frac{1}{2}$, $\lambda = 2$.

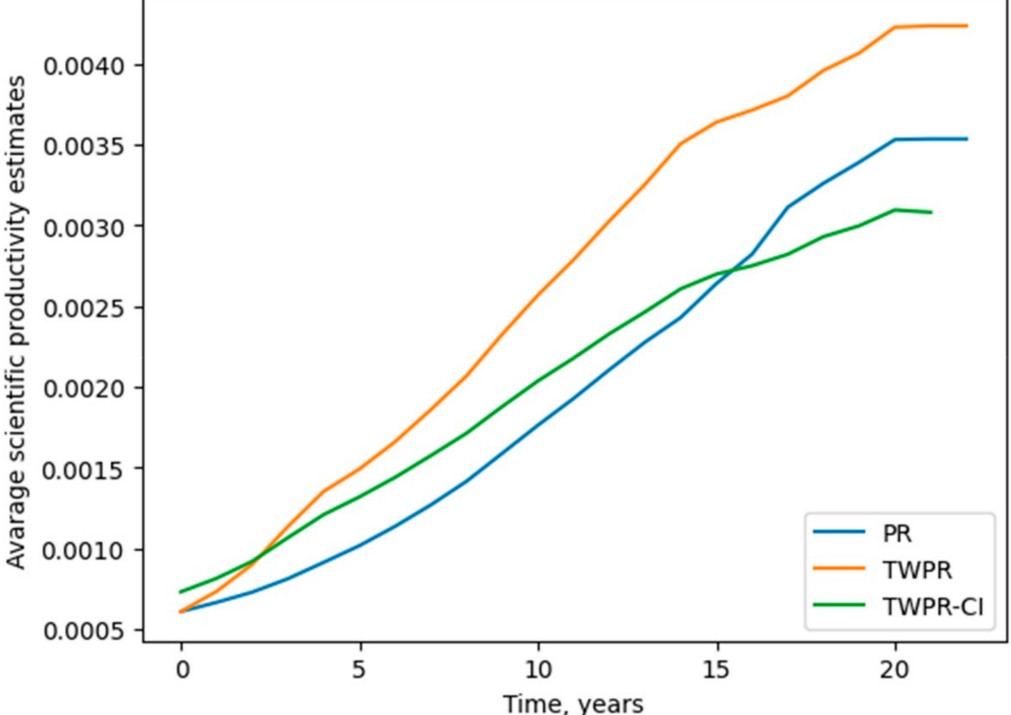

**Figure 5.** A graph comparing the scientific productivity estimates of collective subjects belonging to class O by PR (6), TWPR (6), (8), $\beta = 0$, TWPR-CI (6), (8), $\beta = \frac{1}{2}$, $\lambda = 2$.

## 4. Discussion

### 4.1. Findings

As a result, for collective subjects of class N, estimates of scientific productivity according to the TWPR-CI method are mostly higher at the beginning of observations. These tendencies are observed for the first 10–12 periods following the appearance and citation of

the first scientific publications of the collective subject from class N. The TWPR-CI method ($\beta = \frac{1}{2}$) has a higher sensitivity to assessments of the scientific productivity of new collective subjects. At the same time, the assessment of scientific productivity for other collective subjects (classes WK, O) remains stable. Therefore, the application of the TWPR-CI method makes it possible to increase the sensitivity of the assessment of scientific productivity in comparison with the PR method and the TWPR method for new collective subjects (class N). This tendency is observed until the citation network of the collective subject increases to the appropriate volume. Then, it will have sufficient nodes and connections to use the PR and TWPR methods. For the dataset Citation Network Dataset with the observation period of 2000–2021, the time period when the sensitivity of the TWPR-CI method increases more steeply is, on average, approx. 12 years.

It should be noted that during the citation intensity calculation (7), a significant overestimation of the intensity was found in the first 3–5 periods (years) of observations (Figure 6). This occurs due to the properties of the arctan function. Therefore, it was decided to add a coefficient $\lambda > 1$ to correct this feature. The value was chosen empirically, $\lambda = 2$.

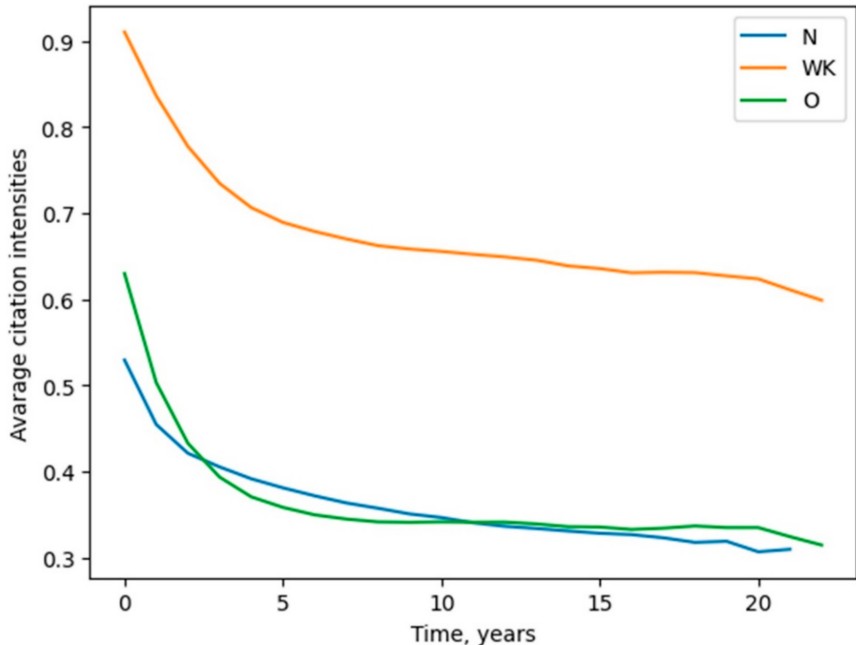

**Figure 6.** Comparison of average citation intensities of collective subjects belonging to classes N, WK, O, $\lambda = 1$.

Any collective entity (classes N, WK, O) is characterized by the accumulation of citations of scientific publications, that is, the growth of their citation network and citation intensity in general (Figure 2). Due to the appearance over time in the citation networks of collective subjects of the WK class of publications that are not cited or rarely cited, the citation intensity curve stabilizes starting from the average of 15 years of observations for $\lambda = 2$.

### 4.2. Limitations and Future Research Lines

An important limitation of the study is that the dataset Citation Network Dataset (ver. 13) has a specific composition. It was established that the majority of scientific publications of the dataset belong to the following scientific areas: computer science, artificial intelligence, artificial neural networks, mathematics, combinatorics, and software engineering. It can be assumed that studying the scientific productivity of collective subjects whose authors work in other areas may lead to slightly different results. This is a separate task for future research. Furthermore, finding the optimal value of the parameter $\beta$ for

the TWPR-CI method is a separate research task. In this implementation, this value was defined at the level of $\beta = \frac{1}{2}$. Accordingly, the impact on the evaluation of the productivity of scientific activity is equally influenced by the intensity of citations and the evaluation of scientific productivity by the TWPR method. Additionally, a separate task of the research is to determine the optimal value $\lambda$ in the equation for calculating the intensity of citations of scientific publications (7).

It should also be noted that the described TWPR-CI method for evaluating the productivity of scientific activity does not provide an opportunity to evaluate the scientific productivity of collective subjects belonging to the NC (non-cited) class. This happens because there is no citation network of scientific publications by authors affiliated with the NC class's collective subjects.

## 5. Conclusions

The study developed the Time-Weighted PageRank method with citation intensity (TWPR-CI) for evaluating the scientific productivity of collective subjects. A dataset of scientific publications, Citation Network Dataset (13 ver.), which is publicly available, was chosen for verification of the method. The dataset contains publications for the period from 1815 to 2021. For analysis, publications that were published for the period from 2000 to 2021 were selected. Four classes of collective subjects were distinguished. For each of these classes, estimates of scientific productivity using the PR (6), TWPR (6), (8), $\beta = 0$ and TWPR-CI (6), (8), $\beta = \frac{1}{2}$, $\lambda = 2$, methods were calculated. An indicator of the intensity of citations of scientific publications (7) was also constructed for $\lambda = 2$. The research hypothesis was confirmed. For collective subjects that belong to class N, estimates of scientific productivity according to the TWPR-CI method are mostly higher at the beginning of observations. Such trends are observed for the first 10–12 periods (years) following the appearance and citation of the first scientific publications belonging to collective subjects from class N. The assessment of scientific productivity for other collective subjects (classes WK, O) at the same time remains stable. This feature allows the TWPR-CI method to be used to evaluate the scientific performance of collective subjects, particularly class N subjects. This is important because using it to evaluate the scientific productivity of such collective subjects (class N) according to the methods of PR and TWPR revealed an underestimation on average during the first ten years of observations. This is due to the small volume of the citation network for new collective subjects. The developed method can help to solve this shortcoming.

Appendix A (Tables A1–A3) presents estimates of the scientific productivity of some collective subjects calculated using the PR, TWPR, and TWPR-CI methods. Ten collective subjects from three classes (WK, O, N) with the highest estimates of scientific productivity were selected.

**Author Contributions:** Conceptualization and methodology, A.K. and Y.A.; software, Y.A.; analysis, A.K., Y.A., S.B. and A.F.; coding, Y.A.; writing—original draft preparation, A.K. and Y.A.; writing—review and editing, A.B., A.K. and. Y.A.; visualization A.K. and. Y.A.; project administration, A.B. All authors have read and agreed to the published version of the manuscript.

**Funding:** This paper was written in the framework of the state order to implement the science program for budget program 217 "Development of Science", IRN No. AP08857218 with the following topic: "Information technology for assessment of scientific activity of universities, research institutes, and their subdivisions".

**Data Availability Statement:** All data are available in this publication. The data used to generate the figures in this article are available in Appendix A. Publicly available datasets analyzed in this study can be found here: Citation Network Dataset: DBLP+Citation, ACM Citation network. (2021). Aminer. Retrieved from: https://www.aminer.org/citation (accessed on 1 August 2022).

**Acknowledgments:** The authors thank the reviewers and editors for their generous and constructive comments that have improved this paper.

**Conflicts of Interest:** The authors declare no conflict of interest.

## Appendix A

**Table A1.** Scientific productivity assessments of collective subjects calculated using the PR method.

| Collective Subjects | Years | | | | | | | | | | | | | | | | | | | | | |
|---|---|---|---|---|---|---|---|---|---|---|---|---|---|---|---|---|---|---|---|---|---|---|
| | 2000 | 2001 | 2002 | 2003 | 2004 | 2005 | 2006 | 2007 | 2008 | 2009 | 2010 | 2011 | 2012 | 2013 | 2014 | 2015 | 2016 | 2017 | 2018 | 2019 | 2020 | 2021 |
| IBM Thomas J, Watson RC, New York | 1.000 | 1.000 | 1.000 | 1.000 | 1.000 | 1.000 | 1.000 | 1.000 | 1.000 | 1.000 | 1.000 | 1.000 | 1.000 | 1.000 | 1.000 | 1.000 | 1.000 | 0.995 | 0.944 | 0.893 | 0.855 | 0.854 |
| Carnegie-Mellon University Pittsburgh, PA | 0.600 | 0.625 | 0.644 | 0.658 | 0.677 | 0.680 | 0.695 | 0.711 | 0.732 | 0.749 | 0.772 | 0.791 | 0.813 | 0.839 | 0.858 | 0.887 | 0.912 | 0.951 | 0.933 | 0.919 | 0.913 | 0.913 |
| Stanford Univ., Stanford, CA | 0.570 | 0.595 | 0.616 | 0.645 | 0.668 | 0.677 | 0.692 | 0.711 | 0.730 | 0.755 | 0.778 | 0.798 | 0.825 | 0.850 | 0.874 | 0.909 | 0.944 | 1.000 | 1.000 | 1.000 | 1.000 | 1.000 |
| Massachusetts Institute of Technology, Cambridge | 0.540 | 0.554 | 0.562 | 0.574 | 0.584 | 0.583 | 0.581 | 0.587 | 0.592 | 0.601 | 0.615 | 0.626 | 0.639 | 0.653 | 0.665 | 0.685 | 0.702 | 0.730 | 0.716 | 0.708 | 0.702 | 0.701 |
| Bell Labs Research, Murray Hill, NJ | 0.490 | 0.501 | 0.510 | 0.510 | 0.502 | 0.478 | 0.459 | 0.442 | 0.423 | 0.408 | 0.392 | 0.375 | 0.361 | 0.349 | 0.343 | 0.335 | 0.330 | 0.320 | 0.300 | 0.281 | 0.268 | 0.267 |
| University of California at Berkeley | 0.420 | 0.438 | 0.464 | 0.503 | 0.557 | 0.593 | 0.635 | 0.669 | 0.683 | 0.700 | 0.720 | 0.730 | 0.749 | 0.769 | 0.788 | 0.816 | 0.841 | 0.880 | 0.865 | 0.852 | 0.837 | 0.837 |
| PARC (Palo Alto Research Center) | 0.290 | 0.294 | 0.302 | 0.305 | 0.310 | 0.301 | 0.289 | 0.278 | 0.268 | 0.259 | 0.249 | 0.240 | 0.234 | 0.229 | 0.225 | 0.220 | 0.215 | 0.208 | 0.191 | 0.175 | 0.163 | 0.163 |
| University of Illinois at Urbana-Champaign, USA | 0.280 | 0.289 | 0.288 | 0.289 | 0.293 | 0.294 | 0.305 | 0.313 | 0.324 | 0.342 | 0.359 | 0.377 | 0.394 | 0.410 | 0.424 | 0.442 | 0.456 | 0.475 | 0.467 | 0.458 | 0.450 | 0.450 |
| Cornell Univ., Ithaca, NY | 0.280 | 0.295 | 0.302 | 0.309 | 0.312 | 0.305 | 0.299 | 0.295 | 0.292 | 0.296 | 0.298 | 0.298 | 0.302 | 0.306 | 0.312 | 0.327 | 0.342 | 0.355 | 0.344 | 0.329 | 0.316 | 0.316 |
| University of Washington | 0.220 | 0.234 | 0.240 | 0.248 | 0.254 | 0.255 | 0.261 | 0.266 | 0.273 | 0.282 | 0.291 | 0.296 | 0.305 | 0.315 | 0.322 | 0.332 | 0.340 | 0.351 | 0.343 | 0.340 | 0.339 | 0.339 |
| Jagiellonian University, Krakow, Poland | 0.010 | 0.012 | 0.015 | 0.016 | 0.019 | 0.021 | 0.024 | 0.026 | 0.028 | 0.030 | 0.033 | 0.035 | 0.037 | 0.038 | 0.040 | 0.042 | 0.043 | 0.045 | 0.044 | 0.044 | 0.044 | 0.044 |
| Aalborg University, Aalborg, Denmark | 0.010 | 0.012 | 0.014 | 0.017 | 0.019 | 0.020 | 0.023 | 0.025 | 0.027 | 0.030 | 0.033 | 0.036 | 0.039 | 0.041 | 0.043 | 0.045 | 0.047 | 0.051 | 0.052 | 0.053 | 0.053 | 0.053 |
| Naval Postgraduate School | 0.010 | 0.010 | 0.011 | 0.011 | 0.010 | 0.010 | 0.010 | 0.010 | 0.010 | 0.010 | 0.010 | 0.011 | 0.011 | 0.011 | 0.012 | 0.012 | 0.012 | 0.013 | 0.012 | 0.012 | 0.011 | 0.011 |
| University of Warwick | 0.010 | 0.010 | 0.011 | 0.011 | 0.012 | 0.013 | 0.013 | 0.013 | 0.013 | 0.014 | 0.014 | 0.015 | 0.015 | 0.016 | 0.017 | 0.018 | 0.019 | 0.020 | 0.020 | 0.021 | 0.022 | 0.022 |
| Lockheed Palo Alto Res. Lab., California, USA | 0.010 | 0.010 | 0.009 | 0.009 | 0.008 | 0.007 | 0.007 | 0.006 | 0.006 | 0.006 | 0.005 | 0.005 | 0.005 | 0.005 | 0.004 | 0.004 | 0.004 | 0.004 | 0.004 | 0.003 | 0.003 | 0.003 |
| Colorado State University, Fort Collins, Colorado | 0.010 | 0.010 | 0.011 | 0.011 | 0.012 | 0.013 | 0.014 | 0.016 | 0.018 | 0.020 | 0.021 | 0.022 | 0.023 | 0.024 | 0.025 | 0.026 | 0.027 | 0.028 | 0.028 | 0.028 | 0.028 | 0.028 |
| University of Kentucky | 0.010 | 0.010 | 0.010 | 0.011 | 0.011 | 0.011 | 0.012 | 0.013 | 0.015 | 0.017 | 0.019 | 0.020 | 0.022 | 0.023 | 0.024 | 0.025 | 0.025 | 0.026 | 0.025 | 0.025 | 0.025 | 0.025 |
| Washington State University | 0.010 | 0.010 | 0.010 | 0.010 | 0.010 | 0.010 | 0.010 | 0.011 | 0.011 | 0.011 | 0.012 | 0.012 | 0.013 | 0.014 | 0.015 | 0.016 | 0.017 | 0.018 | 0.018 | 0.019 | 0.020 | 0.020 |
| ENSICAEN, France | 0.010 | 0.009 | 0.009 | 0.009 | 0.008 | 0.007 | 0.007 | 0.006 | 0.006 | 0.005 | 0.005 | 0.005 | 0.005 | 0.004 | 0.004 | 0.004 | 0.004 | 0.004 | 0.004 | 0.003 | 0.003 | 0.003 |
| University of Connecticut, Farmington, Connecticut | 0.010 | 0.010 | 0.010 | 0.010 | 0.010 | 0.010 | 0.011 | 0.012 | 0.013 | 0.015 | 0.017 | 0.019 | 0.021 | 0.022 | 0.023 | 0.024 | 0.026 | 0.028 | 0.028 | 0.028 | 0.028 | 0.028 |
| Facebook, Menlo Park, CA, USA | 0 | 0 | 0 | 0 | 0 | 0 | 0 | 0 | 0 | 0 | 0.001 | 0.003 | 0.006 | 0.010 | 0.014 | 0.019 | 0.024 | 0.038 | 0.054 | 0.079 | 0.108 | 0.108 |
| DeepMind Technologies Ltd, London, UK | 0 | 0 | 0 | 0 | 0 | 0 | 0 | 0 | 0 | 0 | 0 | 0 | 0 | 0 | 0 | 0 | 0.004 | 0.017 | 0.031 | 0.049 | 0.069 | 0.069 |
| Beijing University of Posts and Tel., Beijing, China | 0 | 0 | 0 | 0 | 0 | 0 | 0 | 0.001 | 0.001 | 0.001 | 0.002 | 0.003 | 0.004 | 0.005 | 0.007 | 0.010 | 0.013 | 0.019 | 0.026 | 0.034 | 0.041 | 0.041 |
| CTO of Baseband SoC at Huawei Technol., Plano, TX | 0 | 0 | 0 | 0 | 0 | 0 | 0 | 0 | 0 | 0 | 0 | 0.001 | 0.001 | 0.002 | 0.003 | 0.005 | 0.008 | 0.013 | 0.019 | 0.024 | 0.028 | 0.028 |
| Aalto University, Espoo, Finland | 0 | 0 | 0 | 0 | 0 | 0 | 0 | 0 | 0 | 0 | 0 | 0.001 | 0.002 | 0.003 | 0.004 | 0.006 | 0.009 | 0.014 | 0.018 | 0.022 | 0.025 | 0.025 |
| Disney Research, Pittsburgh PA | 0 | 0 | 0 | 0 | 0 | 0 | 0 | 0 | 0 | 0 | 0 | 0.001 | 0.002 | 0.004 | 0.006 | 0.008 | 0.010 | 0.015 | 0.017 | 0.020 | 0.021 | 0.021 |
| Max Planck Institute for Intelligent Systems, Germany | 0 | 0 | 0 | 0 | 0 | 0 | 0 | 0 | 0 | 0 | 0 | 0 | 0 | 0.001 | 0.002 | 0.004 | 0.006 | 0.009 | 0.013 | 0.017 | 0.021 | 0.021 |
| Rice Univ. & Max Planck Institute | 0 | 0 | 0 | 0 | 0 | 0 | 0 | 0 | 0.001 | 0.002 | 0.004 | 0.006 | 0.009 | 0.012 | 0.013 | 0.015 | 0.017 | 0.019 | 0.019 | 0.020 | 0.020 | 0.020 |
| Institute for Infocomm Research, A*STAR, Singapore | 0 | 0 | 0 | 0 | 0 | 0 | 0 | 0 | 0.001 | 0.001 | 0.002 | 0.004 | 0.005 | 0.006 | 0.007 | 0.009 | 0.011 | 0.015 | 0.017 | 0.019 | 0.020 | 0.020 |
| University of Lugano, Lugano, Switzerland | 0 | 0 | 0 | 0 | 0 | 0 | 0.001 | 0.001 | 0.002 | 0.002 | 0.003 | 0.005 | 0.007 | 0.009 | 0.010 | 0.012 | 0.013 | 0.015 | 0.017 | 0.019 | 0.020 | 0.020 |

**Table A2.** Scientific productivity assessments of collective subjects calculated using the TWPR method.

| Collective Subjects | Years | | | | | | | | | | | | | | | | | | | | | |
|---|---|---|---|---|---|---|---|---|---|---|---|---|---|---|---|---|---|---|---|---|---|---|
| | 2000 | 2001 | 2002 | 2003 | 2004 | 2005 | 2006 | 2007 | 2008 | 2009 | 2010 | 2011 | 2012 | 2013 | 2014 | 2015 | 2016 | 2017 | 2018 | 2019 | 2020 | 2021 |
| IBM Thomas J, Watson RC, New York | 1.000 | 1.000 | 1.000 | 1.000 | 1.000 | 1.000 | 1.000 | 1.000 | 1.000 | 1.000 | 1.000 | 1.000 | 1.000 | 1.000 | 1.000 | 0.947 | 0.898 | 0.826 | 0.769 | 0.717 | 0.677 | 0.677 |
| Carnegie-Mellon University Pittsburgh, PA | 0.602 | 0.645 | 0.688 | 0.713 | 0.748 | 0.731 | 0.748 | 0.770 | 0.800 | 0.825 | 0.857 | 0.884 | 0.914 | 0.949 | 0.979 | 0.971 | 0.956 | 0.934 | 0.907 | 0.889 | 0.881 | 0.881 |
| Stanford Univ., Stanford, CA | 0.573 | 0.617 | 0.666 | 0.729 | 0.763 | 0.748 | 0.759 | 0.781 | 0.802 | 0.839 | 0.868 | 0.893 | 0.928 | 0.961 | 0.999 | 1.000 | 1.000 | 1.000 | 1.000 | 1.000 | 1.000 | 1.000 |

**Table A2.** *Cont.*

| Collective Subjects | Years | | | | | | | | | | | | | | | | | | | | | |
|---|---|---|---|---|---|---|---|---|---|---|---|---|---|---|---|---|---|---|---|---|---|---|
| | 2000 | 2001 | 2002 | 2003 | 2004 | 2005 | 2006 | 2007 | 2008 | 2009 | 2010 | 2011 | 2012 | 2013 | 2014 | 2015 | 2016 | 2017 | 2018 | 2019 | 2020 | 2021 |
| Massachusetts Institute of Technology, Cambridge | 0.544 | 0.566 | 0.583 | 0.609 | 0.625 | 0.606 | 0.595 | 0.603 | 0.609 | 0.627 | 0.650 | 0.665 | 0.685 | 0.705 | 0.726 | 0.721 | 0.710 | 0.695 | 0.679 | 0.672 | 0.667 | 0.667 |
| Bell Labs Research, Murray Hill, NJ | 0.491 | 0.507 | 0.526 | 0.511 | 0.476 | 0.418 | 0.386 | 0.362 | 0.336 | 0.320 | 0.302 | 0.284 | 0.269 | 0.257 | 0.252 | 0.235 | 0.223 | 0.205 | 0.194 | 0.184 | 0.175 | 0.175 |
| University of California at Berkeley | 0.416 | 0.458 | 0.516 | 0.606 | 0.711 | 0.740 | 0.784 | 0.813 | 0.807 | 0.815 | 0.831 | 0.833 | 0.853 | 0.877 | 0.905 | 0.897 | 0.885 | 0.869 | 0.847 | 0.829 | 0.808 | 0.808 |
| PARC (Palo Alto Research Center) | 0.289 | 0.297 | 0.314 | 0.309 | 0.312 | 0.284 | 0.258 | 0.244 | 0.229 | 0.220 | 0.208 | 0.200 | 0.195 | 0.189 | 0.185 | 0.170 | 0.157 | 0.138 | 0.123 | 0.110 | 0.099 | 0.099 |
| University of Illinois at Urbana-Champaign, USA | 0.284 | 0.294 | 0.295 | 0.297 | 0.307 | 0.308 | 0.329 | 0.342 | 0.358 | 0.388 | 0.413 | 0.440 | 0.463 | 0.486 | 0.506 | 0.505 | 0.499 | 0.482 | 0.468 | 0.452 | 0.441 | 0.441 |
| Cornell Univ., Ithaca, NY | 0.283 | 0.307 | 0.328 | 0.342 | 0.343 | 0.319 | 0.307 | 0.299 | 0.295 | 0.307 | 0.312 | 0.313 | 0.320 | 0.328 | 0.343 | 0.353 | 0.358 | 0.350 | 0.334 | 0.311 | 0.292 | 0.292 |
| University of Washington | 0.223 | 0.243 | 0.259 | 0.279 | 0.287 | 0.279 | 0.284 | 0.291 | 0.300 | 0.313 | 0.325 | 0.331 | 0.345 | 0.357 | 0.369 | 0.363 | 0.355 | 0.341 | 0.330 | 0.327 | 0.328 | 0.328 |
| Jagiellonian University, Krakow, Poland | 0.010 | 0.014 | 0.020 | 0.024 | 0.029 | 0.032 | 0.035 | 0.037 | 0.040 | 0.043 | 0.046 | 0.048 | 0.050 | 0.052 | 0.053 | 0.053 | 0.051 | 0.049 | 0.048 | 0.047 | 0.047 | 0.047 |
| Aalborg University, Aalborg, Denmark | 0.010 | 0.014 | 0.019 | 0.031 | 0.032 | 0.030 | 0.033 | 0.036 | 0.040 | 0.044 | 0.048 | 0.051 | 0.054 | 0.058 | 0.060 | 0.060 | 0.059 | 0.059 | 0.059 | 0.060 | 0.059 | 0.059 |
| Naval Postgraduate School | 0.010 | 0.010 | 0.011 | 0.011 | 0.010 | 0.010 | 0.009 | 0.009 | 0.009 | 0.010 | 0.011 | 0.011 | 0.012 | 0.013 | 0.014 | 0.013 | 0.013 | 0.012 | 0.012 | 0.011 | 0.010 | 0.010 |
| University of Warwick | 0.010 | 0.011 | 0.012 | 0.013 | 0.014 | 0.016 | 0.016 | 0.015 | 0.015 | 0.016 | 0.017 | 0.018 | 0.019 | 0.020 | 0.021 | 0.021 | 0.021 | 0.022 | 0.022 | 0.023 | 0.025 | 0.025 |
| Lockheed Palo Alto Res. Lab., California, USA | 0.010 | 0.009 | 0.008 | 0.007 | 0.006 | 0.005 | 0.004 | 0.004 | 0.004 | 0.004 | 0.003 | 0.003 | 0.003 | 0.003 | 0.003 | 0.003 | 0.003 | 0.002 | 0.002 | 0.002 | 0.002 | 0.002 |
| Colorado State University, Fort Collins, Colorado | 0.010 | 0.011 | 0.012 | 0.013 | 0.014 | 0.017 | 0.019 | 0.021 | 0.025 | 0.028 | 0.028 | 0.029 | 0.030 | 0.030 | 0.032 | 0.032 | 0.031 | 0.030 | 0.030 | 0.030 | 0.029 | 0.029 |
| University of Kentucky | 0.010 | 0.011 | 0.011 | 0.012 | 0.013 | 0.013 | 0.014 | 0.015 | 0.020 | 0.023 | 0.025 | 0.028 | 0.030 | 0.030 | 0.032 | 0.031 | 0.030 | 0.029 | 0.027 | 0.026 | 0.026 | 0.026 |
| Washington State University | 0.010 | 0.010 | 0.010 | 0.010 | 0.012 | 0.011 | 0.012 | 0.012 | 0.013 | 0.013 | 0.014 | 0.014 | 0.016 | 0.017 | 0.018 | 0.019 | 0.020 | 0.020 | 0.021 | 0.022 | 0.023 | 0.023 |
| ENSICAEN, France | 0.010 | 0.009 | 0.008 | 0.007 | 0.006 | 0.005 | 0.005 | 0.004 | 0.004 | 0.004 | 0.003 | 0.003 | 0.003 | 0.003 | 0.003 | 0.003 | 0.002 | 0.002 | 0.002 | 0.001 | 0.001 | 0.001 |
| University of Connecticut, Farmington, Connecticut | 0.010 | 0.010 | 0.010 | 0.011 | 0.010 | 0.011 | 0.014 | 0.015 | 0.017 | 0.020 | 0.024 | 0.026 | 0.029 | 0.030 | 0.031 | 0.032 | 0.033 | 0.032 | 0.032 | 0.032 | 0.031 | 0.031 |
| Facebook, Menlo Park, CA, USA | 0 | 0 | 0 | 0 | 0 | 0 | 0 | 0 | 0 | 0.001 | 0.003 | 0.006 | 0.011 | 0.019 | 0.025 | 0.033 | 0.039 | 0.057 | 0.081 | 0.117 | 0.161 | 0.161 |
| DeepMind Technologies Ltd, London, UK | 0 | 0 | 0 | 0 | 0 | 0 | 0 | 0 | 0 | 0 | 0 | 0 | 0.001 | 0 | 0 | 0.001 | 0.007 | 0.028 | 0.051 | 0.078 | 0.108 | 0.108 |
| Beijing University of Posts and Tel., Beijing, China | 0 | 0 | 0 | 0 | 0 | 0.001 | 0.001 | 0.001 | 0.002 | 0.003 | 0.004 | 0.006 | 0.008 | 0.010 | 0.013 | 0.018 | 0.023 | 0.031 | 0.043 | 0.053 | 0.064 | 0.064 |
| CTO of Baseband SoC at Huawei Technol., Plano, TX | 0 | 0 | 0 | 0 | 0 | 0 | 0 | 0 | 0 | 0 | 0.001 | 0.001 | 0.003 | 0.004 | 0.006 | 0.009 | 0.013 | 0.021 | 0.029 | 0.036 | 0.042 | 0.042 |
| Aalto University, Espoo, Finland | 0 | 0 | 0 | 0 | 0 | 0 | 0 | 0 | 0 | 0 | 0.001 | 0.002 | 0.003 | 0.006 | 0.008 | 0.012 | 0.016 | 0.022 | 0.027 | 0.032 | 0.036 | 0.036 |
| Disney Research, Pittsburgh PA | 0 | 0 | 0 | 0 | 0 | 0 | 0 | 0 | 0 | 0 | 0 | 0.001 | 0.004 | 0.007 | 0.011 | 0.015 | 0.017 | 0.023 | 0.026 | 0.026 | 0.029 | 0.029 |
| Max Planck Institute for Intelligent Systems, Germany | 0 | 0 | 0 | 0 | 0 | 0 | 0 | 0 | 0 | 0 | 0 | 0 | 0.001 | 0.002 | 0.004 | 0.007 | 0.010 | 0.014 | 0.020 | 0.026 | 0.031 | 0.031 |
| Rice Univ. & Max Planck Institute | 0 | 0 | 0 | 0 | 0 | 0 | 0.001 | 0.001 | 0.001 | 0.003 | 0.005 | 0.007 | 0.009 | 0.011 | 0.013 | 0.016 | 0.019 | 0.022 | 0.025 | 0.027 | 0.029 | 0.029 |
| Institute for Infocomm Research, A*STAR, Singapore | 0 | 0 | 0 | 0 | 0 | 0 | 0.001 | 0.001 | 0.001 | 0.003 | 0.005 | 0.007 | 0.009 | 0.011 | 0.013 | 0.016 | 0.019 | 0.022 | 0.025 | 0.027 | 0.029 | 0.029 |
| University of Lugano, Lugano, Switzerland | 0 | 0 | 0 | 0 | 0.001 | 0.001 | 0.001 | 0.002 | 0.003 | 0.004 | 0.006 | 0.009 | 0.012 | 0.015 | 0.017 | 0.019 | 0.020 | 0.022 | 0.023 | 0.024 | 0.026 | 0.026 |

**Table A3.** Scientific productivity assessments of collective subjects calculated using the TWPR-CI method.

| Collective Subjects | Years | | | | | | | | | | | | | | | | | | | | | |
|---|---|---|---|---|---|---|---|---|---|---|---|---|---|---|---|---|---|---|---|---|---|---|
| | 2000 | 2001 | 2002 | 2003 | 2004 | 2005 | 2006 | 2007 | 2008 | 2009 | 2010 | 2011 | 2012 | 2013 | 2014 | 2015 | 2016 | 2017 | 2018 | 2019 | 2020 | 2021 |
| IBM Thomas J, Watson RC, New York | 0.998 | 0.996 | 0.994 | 0.993 | 0.992 | 0.992 | 0.991 | 0.991 | 0.991 | 0.990 | 0.990 | 0.990 | 0.989 | 0.989 | 0.989 | 0.938 | 0.891 | 0.822 | 0.769 | 0.719 | 0.680 | 0.680 |
| Carnegie-Mellon University Pittsburgh, PA | 0.620 | 0.660 | 0.698 | 0.721 | 0.753 | 0.737 | 0.753 | 0.773 | 0.801 | 0.825 | 0.855 | 0.881 | 0.909 | 0.942 | 0.970 | 0.963 | 0.948 | 0.927 | 0.902 | 0.885 | 0.877 | 0.876 |
| Stanford Univ., Stanford, CA | 0.592 | 0.633 | 0.678 | 0.737 | 0.769 | 0.754 | 0.764 | 0.785 | 0.805 | 0.840 | 0.867 | 0.891 | 0.924 | 0.955 | 0.991 | 0.993 | 0.993 | 0.993 | 0.993 | 0.993 | 0.993 | 0.993 |
| Massachusetts Institute of Technology, Cambridge | 0.565 | 0.584 | 0.598 | 0.623 | 0.637 | 0.618 | 0.608 | 0.615 | 0.620 | 0.637 | 0.659 | 0.673 | 0.692 | 0.711 | 0.730 | 0.726 | 0.715 | 0.700 | 0.685 | 0.678 | 0.673 | 0.673 |
| Bell Labs Research, Murray Hill, NJ | 0.514 | 0.527 | 0.543 | 0.528 | 0.494 | 0.439 | 0.408 | 0.384 | 0.360 | 0.344 | 0.327 | 0.309 | 0.295 | 0.283 | 0.279 | 0.262 | 0.250 | 0.233 | 0.223 | 0.212 | 0.204 | 0.203 |
| University of California at Berkeley | 0.444 | 0.482 | 0.536 | 0.621 | 0.720 | 0.747 | 0.789 | 0.817 | 0.811 | 0.819 | 0.834 | 0.835 | 0.854 | 0.877 | 0.903 | 0.895 | 0.884 | 0.869 | 0.847 | 0.830 | 0.810 | 0.810 |
| PARC (Palo Alto Research Center) | 0.324 | 0.330 | 0.345 | 0.340 | 0.343 | 0.316 | 0.291 | 0.278 | 0.263 | 0.255 | 0.243 | 0.235 | 0.230 | 0.224 | 0.220 | 0.206 | 0.193 | 0.175 | 0.161 | 0.148 | 0.138 | 0.137 |

**Table A3.** *Cont.*

| Collective Subjects | Years | | | | | | | | | | | | | | | | | | | | | |
|---|---|---|---|---|---|---|---|---|---|---|---|---|---|---|---|---|---|---|---|---|---|---|
| | 2000 | 2001 | 2002 | 2003 | 2004 | 2005 | 2006 | 2007 | 2008 | 2009 | 2010 | 2011 | 2012 | 2013 | 2014 | 2015 | 2016 | 2017 | 2018 | 2019 | 2020 | 2021 |
| University of Illinois at Urbana-Champaign, USA | 0.317 | 0.325 | 0.323 | 0.323 | 0.331 | 0.331 | 0.350 | 0.362 | 0.377 | 0.406 | 0.430 | 0.455 | 0.477 | 0.499 | 0.518 | 0.517 | 0.511 | 0.495 | 0.482 | 0.467 | 0.456 | 0.456 |
| Cornell Univ., Ithaca, NY | 0.260 | 0.277 | 0.291 | 0.309 | 0.315 | 0.307 | 0.312 | 0.318 | 0.327 | 0.339 | 0.350 | 0.355 | 0.368 | 0.380 | 0.391 | 0.385 | 0.377 | 0.364 | 0.354 | 0.351 | 0.351 | 0.351 |
| University of Washington | 0.255 | 0.281 | 0.319 | 0.358 | 0.391 | 0.395 | 0.403 | 0.412 | 0.419 | 0.428 | 0.442 | 0.455 | 0.469 | 0.487 | 0.500 | 0.494 | 0.484 | 0.475 | 0.468 | 0.471 | 0.471 | 0.471 |
| Jagiellonian University, Krakow, Poland | 0.054 | 0.054 | 0.056 | 0.057 | 0.060 | 0.060 | 0.061 | 0.062 | 0.065 | 0.067 | 0.069 | 0.071 | 0.074 | 0.075 | 0.077 | 0.077 | 0.075 | 0.073 | 0.072 | 0.071 | 0.071 | 0.070 |
| Aalborg University, Aalborg, Denmark | 0.054 | 0.053 | 0.054 | 0.064 | 0.064 | 0.061 | 0.063 | 0.065 | 0.068 | 0.072 | 0.076 | 0.079 | 0.081 | 0.084 | 0.085 | 0.084 | 0.083 | 0.082 | 0.083 | 0.083 | 0.083 | 0.082 |
| Naval Postgraduate School | 0.053 | 0.049 | 0.047 | 0.044 | 0.040 | 0.037 | 0.035 | 0.034 | 0.032 | 0.033 | 0.033 | 0.033 | 0.033 | 0.034 | 0.035 | 0.034 | 0.034 | 0.033 | 0.032 | 0.031 | 0.030 | 0.029 |
| University of Warwick | 0.052 | 0.047 | 0.044 | 0.042 | 0.041 | 0.041 | 0.041 | 0.039 | 0.038 | 0.038 | 0.037 | 0.038 | 0.039 | 0.039 | 0.040 | 0.040 | 0.039 | 0.039 | 0.040 | 0.041 | 0.043 | 0.042 |
| Lockheed Palo Alto Res. Lab., California, USA | 0.036 | 0.026 | 0.023 | 0.025 | 0.027 | 0.028 | 0.027 | 0.029 | 0.032 | 0.036 | 0.040 | 0.047 | 0.053 | 0.059 | 0.064 | 0.066 | 0.069 | 0.071 | 0.070 | 0.067 | 0.064 | 0.064 |
| Colorado State University, Fort Collins, Colorado | 0.052 | 0.047 | 0.044 | 0.042 | 0.042 | 0.043 | 0.046 | 0.048 | 0.052 | 0.055 | 0.056 | 0.057 | 0.057 | 0.057 | 0.058 | 0.058 | 0.057 | 0.056 | 0.056 | 0.056 | 0.055 | 0.054 |
| University of Kentucky | 0.052 | 0.047 | 0.044 | 0.042 | 0.042 | 0.043 | 0.046 | 0.048 | 0.052 | 0.055 | 0.056 | 0.057 | 0.057 | 0.057 | 0.058 | 0.058 | 0.057 | 0.056 | 0.056 | 0.056 | 0.055 | 0.054 |
| Washington State University | 0.052 | 0.047 | 0.042 | 0.039 | 0.039 | 0.037 | 0.036 | 0.037 | 0.037 | 0.037 | 0.037 | 0.038 | 0.039 | 0.041 | 0.043 | 0.044 | 0.044 | 0.045 | 0.047 | 0.048 | 0.048 | 0.048 |
| ENSICAEN, France | 0.057 | 0.054 | 0.050 | 0.049 | 0.047 | 0.044 | 0.044 | 0.042 | 0.0041 | 0.040 | 0.040 | 0.039 | 0.038 | 0.037 | 0.036 | 0.035 | 0.034 | 0.034 | 0.033 | 0.032 | 0.032 | 0.031 |
| University of Connecticut, Farmington, Connecticut | 0.052 | 0.046 | 0.040 | 0.037 | 0.033 | 0.033 | 0.035 | 0.035 | 0.037 | 0.041 | 0.046 | 0.048 | 0.051 | 0.052 | 0.054 | 0.054 | 0.055 | 0.055 | 0.056 | 0.055 | 0.054 | 0.054 |
| Facebook, Menlo Park, CA, USA | 0 | 0 | 0 | 0 | 0 | 0 | 0 | 0 | 0.030 | 0.030 | 0.040 | 0.045 | 0.051 | 0.059 | 0.065 | 0.073 | 0.079 | 0.096 | 0.120 | 0.155 | 0.197 | 0.197 |
| DeepMind Technologies Ltd, London, UK | 0 | 0 | 0 | 0 | 0 | 0 | 0 | 0 | 0 | 0 | 0 | 0 | 0.047 | 0.046 | 0.039 | 0.032 | 0.050 | 0.075 | 0.095 | 0.122 | 0.150 | 0.151 |
| Beijing University of Posts and Tel., Beijing, China | 0 | 0 | 0 | 0 | 0.009 | 0.010 | 0.007 | 0.007 | 0.008 | 0.010 | 0.012 | 0.015 | 0.018 | 0.021 | 0.025 | 0.031 | 0.036 | 0.046 | 0.060 | 0.071 | 0.083 | 0.083 |
| CTO of Baseband SoC at Huawei Technol., Plano, TX | 0 | 0 | 0 | 0 | 0 | 0 | 0 | 0 | 0.007 | 0.011 | 0.014 | 0.016 | 0.019 | 0.022 | 0.025 | 0.028 | 0.034 | 0.045 | 0.055 | 0.063 | 0.069 | 0.068 |
| Aalto University, Espoo, Finland | 0 | 0 | 0 | 0 | 0 | 0 | 0 | 0 | 0.031 | 0.035 | 0.010 | 0.013 | 0.017 | 0.021 | 0.025 | 0.030 | 0.036 | 0.045 | 0.051 | 0.057 | 0.061 | 0.060 |
| Disney Research, Pittsburgh PA | 0 | 0 | 0 | 0 | 0 | 0 | 0 | 0 | 0 | 0.008 | 0.010 | 0.028 | 0.036 | 0.042 | 0.048 | 0.052 | 0.056 | 0.062 | 0.066 | 0.068 | 0.070 | 0.069 |
| Max Planck Institute for Intelligent Systems, Germany | 0 | 0 | 0 | 0 | 0 | 0 | 0 | 0 | 0 | 0 | 0.031 | 0.019 | 0.021 | 0.027 | 0.033 | 0.040 | 0.046 | 0.052 | 0.060 | 0.067 | 0.072 | 0.072 |
| Rice Univ. & Max Planck Institute | 0 | 0 | 0 | 0 | 0 | 0 | 0.037 | 0.046 | 0.046 | 0.049 | 0.053 | 0.056 | 0.061 | 0.064 | 0.066 | 0.067 | 0.068 | 0.068 | 0.067 | 0.066 | 0.065 | 0.065 |
| Institute for Infocomm Research, A*STAR, Singapore | 0 | 0 | 0 | 0 | 0 | 0 | 0.014 | 0.014 | 0.015 | 0.019 | 0.023 | 0.027 | 0.031 | 0.034 | 0.036 | 0.038 | 0.043 | 0.048 | 0.053 | 0.055 | 0.058 | 0.057 |
| University of Lugano, Lugano, Switzerland | 0 | 0 | 0 | 0 | 0.035 | 0.025 | 0.022 | 0.022 | 0.022 | 0.023 | 0.026 | 0.030 | 0.035 | 0.039 | 0.041 | 0.045 | 0.046 | 0.048 | 0.051 | 0.053 | 0.054 | 0.054 |

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
