# Peer review of "The Scientific Productivity of Collective Subjects Based on the Time-Weighted PageRank Method with Citation Intensity"

_publications, doi:10.3390/publications10040040_

Round 1

Reviewer 1 Report

The article is an excellent contribution to the issue of evaluating scientific production with an emphasis on evaluating new collective subjects .

Minor comments are related to the formal arrangement. First of all, it is necessary to follow the structure and names of the sections: Introduction, Materials and Methods, Results, Discussion, Conclusions. It is convenient to further divide these sections into chapters. Please check the numbering, section 4 is missing in the text, section 5 appears twice.

The explanation of basic terms is complete in the article, but it is not found in one place. I suggest considering a separate subsection in the Materials and methods section, dedicated to the explanation of the basic terms used in the article (collective and individual subjects, categories of subjects...)

In the future, it would be useful to process, for example, national data from one of the publication and citation databases (WoS, SCOPUS, some national database etc.) in this way.

Author Response

Good afternoon, dear reviewer. We are very grateful for your comments. They will certainly improve the quality of the article.  We have done corrections according all your comments: 1. The numbering of sections has been corrected. The Literature review section is included in the Introduction section. 2. The main definitions were transferred to section 2.1. 3. Your recommendations will be taken into account in the further researches.
Thank you. Have a nice day

Reviewer 2 Report

The article “The scientific productivity of collective subjects based on the Time-Weighted PageRank method with citation intensity” is a study investigating the use of modified PageRank methods that take into account the ageing of citations and their intensity in the process of ranking research institutions. A specific novel method called TWPR-CI is proposed, formally defined and described, and tested on a huge data set of over five million publications and almost 50 million citations. The results of the ranking (evaluation) process applied to a time range of more than 20 years are then compared with two other PageRank-based techniques and it is concluded that the novel method has some advantages over the established ones.

While the paper is generally well written, the idea is original, the amount of the inspected data impressive, and the results promising, there are some issues that definitely need to be addressed and they are highlighted in the following paragraphs.

What is the rationale for using arctg in the formula on line 281? This should be explained in more detail.

Regarding Figure 1 on p. 8, why is it not in colour like the other figures? By the way, the charts in all the other figures (Figure 2 – Figure 6) have no axis legends at all. This is really annoying. Please improve it.

As to “some collective subjects” on line 486: how were they chosen? Some description should be provided here. Also, in the corresponding tables A1, A2, and A3 in the appendix there seem to be three groups of collective subjects probably representing the three different classes mentioned earlier in the manuscript, but no explanation whatsoever is given in the table caption. This must be amended. In addition, in Table A1 in the column for year 2000 the figures have only two decimal numbers unlike all others. Why is that?

When you mention “sensitivity” on line 189, it would be good to provide a definition of it. What do you really understand by sensitivity? And likewise, when you talk about “intensity”, it would be great to explain what exactly you mean by that (provide a definition), in Section 3 on methods and data at the latest.

Please check all formulas again (not only the numbered ones but also in-text formulas) and make sure that all parameters are explained. For example, it appears that parameters g and s on line 268 are undefined.

The term “decrease” on line 391 is not true. Please correct it. Also, “increases” on line 393 is imprecise – it actually increases more steeply. Please improve this.

The sentence starting with “The publication” on line 253 and ending with “coefficient” on line 254 does not make sense. Please revise it.

When mentioning “aging of citations” on line 174, you might consider referring to doi:10.12700/APH.12.6.2015.6.9 for related work.

The phrase “is the coefficient” on line 288 looks strange and incomplete. Which coefficient? Please improve this expression.

References in text should be in square brackets, e.g. “[6]” and “[6, 8]” on lines 341, 342, and many others. Please check them and correct them.

Lines 236, 247, and 388 begin with a comma or a full stop. Please improve that.

Please replace:

·       “possibilities the PageRank method” with “possibilities of the PageRank method” on line 116,

·       “Scimago magazine Journal Ranking” with “SCImago Journal Ranking” on line 131,

·       “of scientific pi is determined” with “of scientific publication pi is determined” on line 258,

·       “authors, which are affiliated” with “authors, who are affiliated” on line 287,

·       “is the moment from which the calculation of the intensity of citations of scientific publications for the collective subject Uh.” with “is the moment from which the calculation of the intensity of citations of scientific publications for the collective subject Uh begins.” on line 288,

·       “5354309 scientific publications as well as 48227950 citations” with
“5,354,309 scientific publications as well as 48,227,950 citations” on line 317,

·       “As can be seen from fig. 3 estimates” with “As can be seen from Figure 3 estimates” on line 386,

·       “Figure 3 shows changes” with “Figure 4 shows changes” on line 397,

·       “As can be seen from fig. 4 estimates” with “As can be seen from Figure 4 estimates” on line 400,

·       “In fig. 5. shows the superiority” with “Figure 5 shows the superiority” on line 414,

·       “(Fig. 6)” with “(Figure 6)” on line 433,

·       “(Fig. 2)” with “(Figure 2)” on line 440,

·       “belonging to the NC class” with “belonging to the NC (non-cited) class” on line 460, and

·       “available at Appendix A” with “available in Appendix A” on line 496.

Author Response

Good afternoon, dear reviewer. We are very grateful for your comments. They will certainly improve the quality of the article. We have done corrections according all your comments: 1. Selection of the arctangent function has been explained (line 358). 2. The axes in Figures 2-6 have been labeled. Figure 1 was generated in “black and white” using Gephi 0.9.7 software . The color in this drawing will have no additional meaning. 3. Changes to table A1 were made. Tables A1-A3 show 10 collective subjects from each class. And the calculation methods used is presented in the table headers. The choice of collective subjects is described (line 700). 4. Definitions «intensity» and «sensitivity» have been introduced (line 251). The main definitions were transferred to section 2.1. 5. The typo in formula (6) was corrected. 6. The term "decrease" has been changed. (line 638). 7. The article (doi:10.12700/APH.12.6.2015.6.9) has been added to the references. 8. Also (6), (8) are references to formulas in this work, not to articles (brackets, not square brackets). 9. The rest of your comments were taken into account and the text was amended.
Thank you. Have a nice day!

Round 2

Reviewer 2 Report

The authors have properly revised the manuscript and addressed all the concerns I raised. I have no objections now.